

# Efficient sampling of shiitake-inoculated oak logs to determine the log-to-mushroom transfer factor of stable cesium

Martin O'Brien[1], Masakazu Hiraide[2], Yoshimi Ohmae[1], Naoto Nihei[1], Satoru Miura[2] and Keitaro Tanoi[1]

[1] Graduate School of Agricultural and Life Sciences, The University of Tokyo, Tokyo, Japan
[2] Forestry and Forest Products Research Institute, Tsukuba, Ibaraki, Japan

## ABSTRACT

**Background:** Stable cesium ($^{133}$Cs) naturally exists in the environment whereas recently deposited radionuclides (e.g., $^{137}$Cs) are not at equilibrium. Stable cesium has been used to understand the long-term behavior of radionuclides in plants, trees and mushrooms. We are interested in using $^{133}$Cs to predict the future transfer factor (TF) of radiocesium from contaminated logs to shiitake mushrooms in Eastern Japan. However, the current methodology to obtain a representative wood sample for $^{133}$Cs analysis involves mechanically breaking and milling the entire log (excluding bark) to a powder prior to analysis. In the current study, we investigated if sawdust obtained from cutting a log along its length at eight points is as robust but a faster alternative to provide a representative wood sample to determine the TF of $^{133}$Cs between logs and shiitake.

**Methods:** Oak logs with ready-to-harvest shiitake fruiting bodies were cut into nine 10-cm discs and each disc was separated into bark, sapwood and heartwood and the concentration of $^{133}$Cs was measured in the bark, sapwood, heartwood, sawdust (generated from cutting each disc) and fruiting bodies (collected separately from each disc), and the wood-to-shiitake TF was calculated.

**Results:** We found that the sawdust-to-shiitake TF of $^{133}$Cs did not differ ($P = 0.223$) compared to either the sapwood-to-shiitake TF or heartwood-to-shiitake TF, but bark did have a higher concentration of $^{133}$Cs ($P < 0.05$) compared to sapwood and heartwood. Stable cesium concentration in sawdust and fruiting bodies collected along the length of the logs did not differ ($P > 0.05$).

**Discussion:** Sawdust can be used as an alternative to determine the log-to-shiitake TF of $^{133}$Cs. To satisfy the goals of different studies and professionals, we have described two sampling methodologies (Methods I and II) in this paper. In Method I, a composite of eight sawdust samples collected from a log can be used to provide a representative whole-log sample (i.e., wood and bark), whereas Method II allows for the simultaneous sampling of two sets of sawdust samples—one set representing the whole log and the other representing wood only. Both methodologies can greatly reduce the time required for sample collection and preparation.

Corresponding author
Martin O'Brien,
martinobrien009@gmail.com

## INTRODUCTION

Over 25 million logs are used annually for shiitake mushroom (*Lentinula edodes*) production in Japan (*Ministry of Agriculture, Forestry & Fisheries, 2017*). Prior to the nuclear reactor accident at the Fukushima Daiichi Nuclear Power Plant in 2011, Fukushima prefecture was a major source of trees used for shiitake log-cultivation (*Miura, 2016*). As a result of the accident, the Japanese government restricted radiocesium ($^{134}$Cs and $^{137}$Cs) in general foods, including mushrooms cultivated on logs, to ≤ 100 Bq/kg fresh weight (FW) (*Ministry of Health, Labour & Welfare, 2012*). The transfer of radiocesium from fallout-contaminated logs to shiitake, known as the transfer factor (TF), was determined to be 0.43, with 90% of samples having a TF of less than 2 (*Forestry Agency, 2012*). A provisional limit of radiocesium allowed in logs for mushroom cultivation was then determined to be 50 Bq/kg FW (i.e., radiocesium in logs (Bq/kg) = 100 Bq/kg FW/2) (*Ministry of Agriculture, Forestry & Fisheries, 2012*). Because shiitake mycelia colonizes sapwood (*Tokimoto, 2005*), it is presumed to obtain the majority of its nutrients exclusively from this part of a log. However, the most contaminated part of a log immediately after the accident was the bark and not the inner wood (*Kuroda, Kagawa & Tonosaki, 2013*). In the coming years, it is believed that radiocesium will be absorbed through the bark into the wood of trees (*Mahara et al., 2014*; *Wang et al., 2018*) and be taken up through the roots from contaminated soil (*IAEA, 2006*) giving rise to higher concentrations of radiocesium within logs; evidence is emerging that the contribution from root uptake may be more significant than bark absorption (*Ohashi et al., 2017*). It has been modeled that the maximum $^{137}$Cs concentration in xylem wood (120 Bq/kg) will occur in 2039, compared with 6–47 Bq/kg found in contaminated oak sapwood in 2012 (*Mahara et al., 2014*). Although cultivating shiitake from oak logs grown in contaminated regions of Eastern Japan is currently not recommended, the forestry industry needs more clarity about the long-term trend of radiocesium transfer from contaminated logs to shiitake.

The consumption of food contaminated with radionuclides is a significant route of radionuclide intake by humans (*ICRP, 1993*). Although the current study is related to the long-term behavior of radiocesium in forests and forest products, readers interested in the on-going monitoring of mushrooms in Japan should consult *Merz, Shozugawa & Steinhauser (2015)*, *Tagami, Uchida & Ishii (2017)*, *Orita et al. (2017)* and *Prand-Stritzko & Steinhauser (2018)*.

Stable cesium originates from the mineral component of soil, and therefore naturally exists within trees (*Mahara et al., 2014*), and can be used to understand the long-term behavior of radiocesium (*Rühm et al., 1999*; *Yoshida et al., 2004*; *Karadeniz & Yaprak, 2007*). However, a more efficient sampling method is required to obtain a representative wood sample for $^{133}$Cs analysis because the current in-house methodology is both time and labor intensive (Fig. 1). By cutting a log at multiple points along its length and collecting the sawdust for $^{133}$Cs analysis would greatly facilitate the sampling of a larger number of logs; a method involving fewer steps would also minimize cross-contamination (*Keith et al., 1983*). Sawdust samples have been used previously to measure $^{133}$Cs

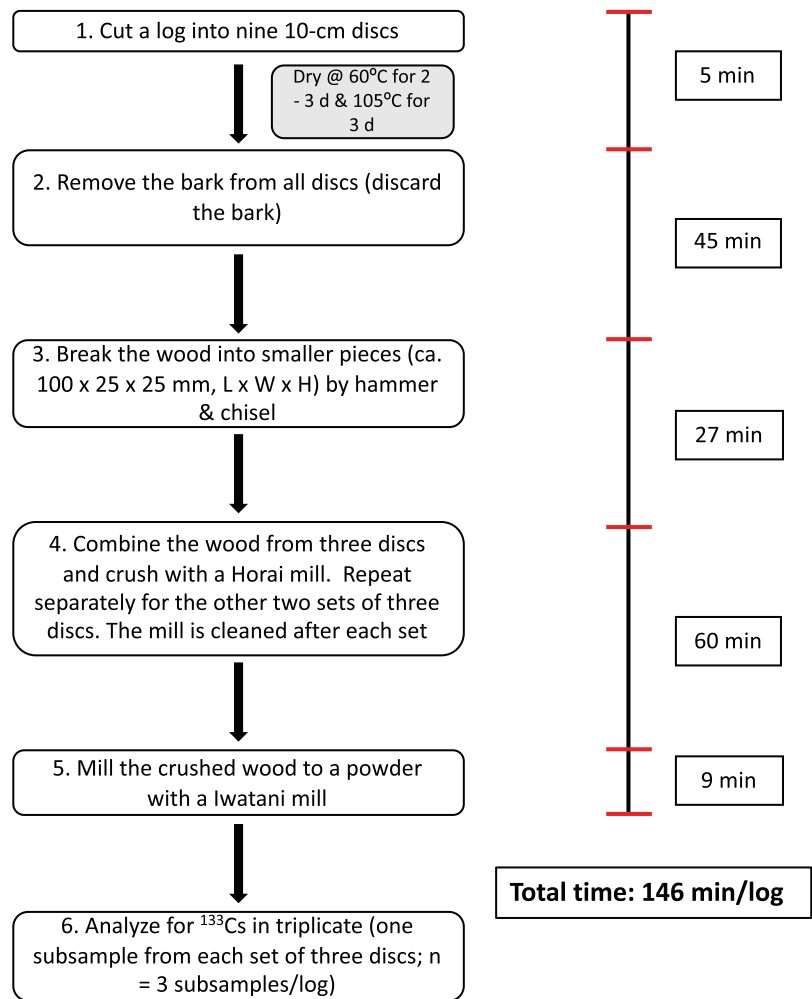

**Figure 1 Sampling and preparing a representative wood sample for $^{133}$Cs analysis—an in-house sampling method.** All fruiting bodies were collected prior to cutting the log. The approximate time required for one technician to collect a wood sample (in triplicate) from one log (size = 20 × 90 cm) is shown on the right; the total time stated excludes drying the wood and collecting and processing fruiting bodies.                                                           

(*Yoshida et al., 2004*) and radiocesium concentrations in standing trees (*Yoshida et al., 2004*; *Schell et al., 1996*).

Our goal in this study was threefold. First, we tested if the sawdust-to-shiitake TF of $^{133}$Cs was comparable to the sapwood-to-shiitake and heartwood-to-shiitake TF. Second, if differences were observed between wood and sawdust TFs, we wanted to know if it was due to a heterogeneous distribution of $^{133}$Cs across heartwood, sapwood and bark. As $^{133}$Cs is not an essential element for plant growth (*Taiz et al., 2015*), little is known about its distribution within trees. However, because of the similar chemical properties of cesium and potassium, it is believed $^{133}$Cs could be translocated to the growing tissues of a tree (*Wyttenbachl et al., 1995*) such as the bark. A higher concentration of $^{133}$Cs in the bark (i.e., the denominator in the equation to calculate the TF) would result in an underestimation of the sawdust-to-shiitake TF. Third, we determined the $^{133}$Cs

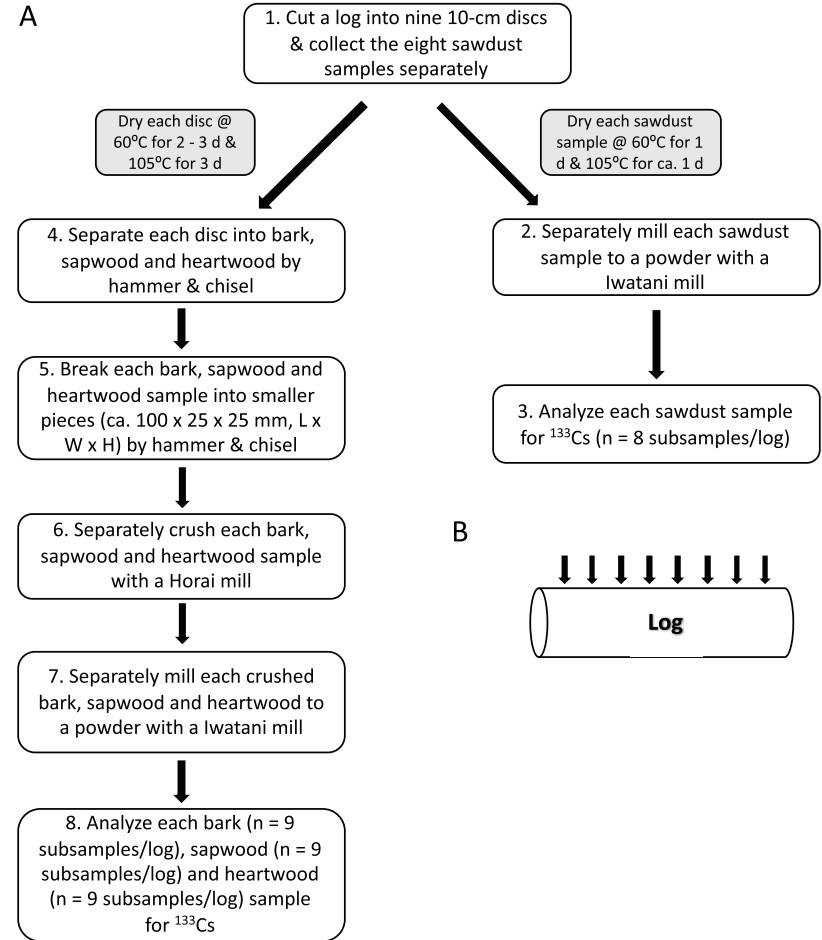

**Figure 2 Sampling and preparing bark, wood and sawdust samples for $^{133}$Cs analysis.** (A) The method used in the current study. (B) Locations where logs were cut. All fruiting bodies were collected prior to cutting the log.

concentration in sawdust and fruiting bodies along the length of the logs. The distribution of $^{133}$Cs will have implications for the number of sawdust and fruiting body samples needed to represent the entire log and mushroom crop, respectively. Based on the findings of this study, we propose two different sampling methodologies using sawdust to determine the log-to-shiitake TF of $^{133}$Cs.

# MATERIALS AND METHODS

## Experimental design

Ten oak logs and their ready-to-eat shiitake fruiting bodies were sampled. Each log was cut at *ca.* 10-cm intervals along its length ($n = 9$ discs/log). The sawdust produced from cutting each disc ($n = 8$ sawdust samples/log) and fruiting bodies present on each disc ($n = 9$ fruiting body samples/log) were collected and analyzed for $^{133}$Cs concentration. For five of the 10 logs, each 10-cm disc was separated into bark ($n = 9$ samples/log), sapwood ($n = 9$ samples/log) and heartwood ($n = 9$ samples/log) and also analyzed for $^{133}$Cs concentration. An overview of the sampling method is shown in Fig. 2.

## Oak logs and shiitake mushroom samples

*Quercus serrata* "konara" is the most common oak species grown in Eastern Japan and commonly used for shiitake cultivation in Japan. Logs with shiitake mushrooms were supplied by a grower located in Ibaraki Prefecture, Japan. This grower purchases up to 200,000 logs per year from oak suppliers throughout Japan. The grower reported that the logs used in the current study were sourced from trees felled in either December 2016 or January 2017 and allowed to dry for 1 month before inoculation. Logs were inoculated with multiple 2-cm plug spawns in either January or February 2017. Fruiting occurred in a humidity- and temperature-controlled environment with logs stored in a horizontal orientation. To minimize the influence of possible confounding factors affecting $^{133}$Cs distribution in logs and fruiting bodies, all logs used in this study were (1) from the same species of oak (*Q. serrata*), (2) sourced from the same region (Sano-city, Tochigi Prefecture), (3) inoculated with the same strain of shiitake (F103; Fujishukin Co. Ltd., Minami-arupusu, Japan), (4) managed similarly from inoculation to fruiting, and (5) producing their first crop of fruiting bodies which were ready to collect and eat on the day of sampling. Selected logs needed to have at least one fruiting body present on each 10-cm interval along its length.

## Collecting logs and shiitake fruiting bodies

Logs and fruiting bodies were collected in March 2018. For each log, 10-cm intervals along its length were demarcated and labeled. Fruiting bodies growing on each 10-cm disc were collected, the bottom 1 cm of the stipe was removed with a knife to eliminate any attached pieces of bark and then the fruiting bodies were placed into factory-new paper bags. When sampling was complete, logs and fruiting bodies were transported to the laboratory.

## Sample preparation
### Fruiting bodies

Fruiting bodies from each 10-cm disc were weight and thinly sliced (*ca.* 2–3 mm) before being placed loosely on a polypropylene mesh, dried overnight at 60 °C and then at 105 °C for 1 day and reweighed to determine the dry weight (DW). The use of a mesh avoided the fruiting bodies adhering to the inside of the paper bags while drying and the initial drying at 60 °C ensured the sliced fruiting bodies were not too hard for milling. The dried fruiting bodies were milled using an Iwatani mill to a powder and placed into air-tight factory-new plastic bags which in turn were placed into a larger air-tight plastic bag containing silica gel sachets.

### Sawdust, wood and bark

The weight, length and diameter of each log was recorded. Discs were cut from each log using a circular saw (model LS1500; Makita, Anjo, Japan) and the sawdust was collected using a dust collector connected to a vacuum. The wet and DW of each sawdust, wood and bark samples were recorded and their DW determined. Each sawdust sample was placed into a separate pre-labelled factory-new plastic container, dried for 1 day at

60 °C and at 105 °C until the DW became constant (*ca.* 1 day). To determine DW of wood and bark, each 10-cm disc was first dried at 60 °C for 2–3 days and then at 105 °C until the weight became constant (*ca.* 3 days), and then each disc was separated into the bark, sapwood and heartwood parts using a hammer and chisel. We distinguished the heartwood from the sapwood by the darker color of the heartwood. Bark, sapwood and heartwood samples were crushed using Horai V-360 mill (Japan) and all parts (including sawdust) were milled to a powder using an Iwatani mill. Samples were placed into air-tight factory-new plastic bags which were then placed into a larger air-tight plastic bag containing silica gel sachets.

### $^{133}$Cs analysis

To keep the number of samples manageable and to focus our sampling efforts where variation in $^{133}$Cs concentration was likely to be high (i.e., within logs) rather than where it was likely to be low (i.e., between analytical replicates of individual samples), multiple wood, bark and fruiting body samples were collected from each log and no analytical replicates were used. To confirm the soundness of this decision, five randomly selected samples representing the materials in this study were digested in duplicate and the $^{133}$Cs concentration measured in each. The deviation in $^{133}$Cs concentration between these duplicate digested samples was very low compared to the variation observed within logs (see Table 1); for example, the mean (± standard deviation (SD)) concentration of $^{133}$Cs in the duplicate heartwood, sapwood, bark, sawdust and fruiting body samples were 12 ± 0.2, 25 ± 0.4, 41 ± 2.0, 23 ± 0.1 and 433 ± 1.4 µg/kg DW, respectively.

Fruiting bodies, log parts and sawdust samples (*ca.* 0.3 g) were digested in 10 ml of $HNO_3$ (60–61% (mass/mass); Wako, Japan) using the Multiwave 3000 digester (PerkinElmer Anton Paar, Austria); samples were digested at 600 W for 40 min at ≥120 °C (30 bars of pressure). The digestates were diluted 1-in-5 with Milli-Q water (Merck, Darmstadt, Germany) and filtered (Advantec 0.2 µm cellulose acetate filters; Toyo Roshi Kaisha Ltd, Tokyo, Japan) prior to $^{133}$Cs analysis. The $^{133}$Cs content was measured by inductively coupled plasma-mass spectrometry (ICP-MS, NexION 350S; PerkinElmer, Waltham, MA, USA) with cesium standard solution (998 mg/l; Wako, Japan) and indium (two µg/l) as the internal standard (1,000 mg/l; Wako, Japan). To minimize variation between ICP-MS runs, wood, bark, sawdust and fruiting bodies of individual logs were analyzed in the same run. At the time of ICP-MS analysis, a subsample (0.5 g) of each sample was oven dried at 105 °C for 48 h for residual moisture determination. For quality control purposes, the National Institute of Standards and Technology standard reference material 1575a (Pine needles) was included in each analysis. Average precision for $^{133}$Cs was 8.5%. Stable cesium values were determined on a DW basis unless stated otherwise.

### Data analysis

The heartwood-to-shiitake and sapwood-to-shiitake TFs were determined between each disc and the fruiting bodies growing on that disc (Eq. (1)), unless stated otherwise.

$$\text{Transfer factor} = \frac{^{133}\text{Cs concentration in fruiting bodies}}{^{133}\text{Cs concentration in heartwood or sapwood}} \qquad (1)$$

Table 1 Transfer factors (TF) of $^{133}$Cs and $^{133}$Cs concentrations in oak log parts, sawdust and shiitake fruiting bodies ($n$ = 5 or 10 logs).

| Log ID | Transfer factor[1] | | | $^{133}$Cs concentration (µg/kg dry weight) | | |
|---|---|---|---|---|---|---|
| | Mean | SD | Range | Mean | SD | Range |
| Heartwood ($n$ = 9 discs/log) | | | | | | |
| 1 | 40 | 10.4 | 20–55 | 20 | 6.1 | 13–33 |
| 2 | 23 | 4.8 | 13–28 | 60 | 25.4 | 39–122 |
| 3 | 24 | 8.4 | 15–43 | 28 | 6.7 | 20–39 |
| 4 | 32 | 9.7 | 13–48 | 14 | 6.3 | 8–29 |
| 5 | 30 | 7.5 | 20–42 | 13 | 2.5 | 10–17 |
| Mean ± SD[2] | 29[a] ± 6.9 | | | 27[a] ± 19.3 | | |
| Sapwood ($n$ = 9 discs/log) | | | | | | |
| 1 | 31 | 6.6 | 20–41 | 25 | 4.8 | 18–33 |
| 2 | 22 | 5.8 | 13–29 | 62 | 28.1 | 41–127 |
| 3 | 25 | 7.6 | 18–43 | 25 | 4.4 | 21–33 |
| 4 | 29 | 6.0 | 18–36 | 14 | 4.2 | 10–24 |
| 5 | 27 | 7.5 | 16–38 | 15 | 4.0 | 10–23 |
| Mean ± SD[2] | 27[a] ± 3.3 | | | 28[a] ± 19.5 | | |
| Bark ($n$ = 9 discs/log) | | | | | | |
| 1 | – | – | – | 79 | 11.6 | 65–102 |
| 2 | – | – | – | 125 | 49.2 | 94–251 |
| 3 | – | – | – | 58 | 10.3 | 42–74 |
| 4 | – | – | – | 43 | 6.3 | 34–52 |
| 5 | – | – | – | 51 | 6.3 | 42–61 |
| Mean ± SD[2] | – | | | 71[b] ± 32.7 | | |
| Sawdust ($n$ = 8 cuts/log) | | | | | | |
| 1 | 27 | 3.7 | 22–31 | 27 | 2.9 | 24–31 |
| 2 | 22 | 3.4 | 16–26 | 57 | 15.0 | 47–93 |
| 3 | 24 | 3.9 | 18–30 | 26 | 2.4 | 22–30 |
| 4 | 26 | 2.9 | 22–31 | 14 | 1.8 | 12–18 |
| 5 | 19 | 4.9 | 14–29 | 21 | 4.5 | 14–28 |
| 6 | 21 | 6.8 | 16–37 | 37 | 7.9 | 20–46 |
| 7 | 27 | 5.7 | 21–37 | 13 | 2.8 | 11–17 |
| 8 | 26 | 7.8 | 14–38 | 11 | 3.5 | 6–15 |
| 9 | 23 | 3.8 | 21–29 | 37 | 4.3 | 30–44 |
| 10 | 15 | 2.8 | 11–20 | 20 | 3.6 | 13–24 |
| Mean ± SD[2] (logs 1–5) | 24[a] ± 3.3 | | | 29 ± 16.2 | | |
| Mean ± SD[3] (logs 1–10) | 23 ± 4.0 | | | 26 ± 14.1 | | |
| Fruiting bodies ($n$ = 9 discs/log) | | | | | | |
| 1 | – | – | – | 743 | 111.0 | 612–1,000 |
| 2 | – | – | – | 1,255 | 160.1 | 1,092–1,620 |
| 3 | – | – | – | 617 | 104.9 | 539–887 |
| 4 | – | – | – | 385 | 61.4 | 267–458 |
| 5 | – | – | – | 383 | 30.5 | 327–427 |

(Continued)

| Log ID | Transfer factor[1] | | | [133]Cs concentration (μg/kg dry weight) | | |
|---|---|---|---|---|---|---|
| | Mean | SD | Range | Mean | SD | Range |
| 6 | – | – | – | 744 | 32.7 | 703–791 |
| 7 | – | – | – | 347 | 32.3 | 310–412 |
| 8 | – | – | – | 258 | 37.6 | 211–324 |
| 9 | – | – | – | 886 | 123.5 | 689–1,086 |
| 10 | – | – | – | 297 | 32.5 | 251–344 |
| Mean ± SD[2] (logs 1–5) | – | – | – | 677 ± 358.4 | | |
| Mean ± SD (logs 1–10) | – | – | – | 591 ± 319.3 | | |
| SEM[4] | 1.3 | | | 8.1 | | |
| Significance | 0.223 | | | 0.023 | | |

**Notes:**
[1] TF values were calculated using [133]Cs data expressed on a dry weight basis.
[2] $n$ = 5 logs.
[3] The sawdust-to-shiitake TF was 4.2 when expressed on a fresh weight basis.
[4] Standard error of the mean.
Mean values in columns in bold font with a common superscript letter do not differ significantly as determined with the Tukey post hoc test. Mean values in each column without a superscript letter were excluded from the statistical analysis.

The average [133]Cs concentration in fruiting bodies growing on each pair of discs that produced a sawdust sample was used to calculate the sawdust-to-shiitake TF (Eq. (2)).

$$\text{Transfer factor} = \frac{\text{Average } ^{133}\text{Cs concentration in fruiting bodies from disc pair}}{^{133}\text{Cs concentration in sawdust from disc pair}} \quad (2)$$

A one-way ANOVA followed by the Tukey post hoc test was used to compare TF values based on heartwood, sapwood and sawdust. A similar analysis was used to compare [133]Cs concentration between heartwood, sapwood and bark and, between fruiting bodies growing on each disc and cuts containing sawdust along the length of the logs.

To demonstrate that collecting fruiting bodies from four of the nine discs (per log) would be representative of the shiitake crop on a log, [133]Cs concentration in fruiting bodies growing on odd- (i.e., 1, 3, 5, 7, 9) and even-numbered discs (i.e., 2, 4, 6, 8) were compared with the paired sample $t$-test. To demonstrate that the whole-log TF and wood-only TF can be determined simultaneously, the TF based on sawdust collected from odd-numbered cuts (i.e., 1, 3, 5, 7) was compared with sawdust from even-numbered cuts (i.e., 2, 4, 6, 8) (Eq. (3)) using the paired sample $t$-test.

$$\text{Transfer factor} = \frac{\text{Average } ^{133}\text{Cs concentration in fruiting bodies from a log}}{\text{Average } ^{133}\text{Cs concentration in odd- or even-numbered sawdust samples}} \quad (3)$$

Correlation coefficients of [133]Cs concentrations in log parts, sawdust and shiitake fruiting bodies and their level of significance were determined. All statistical analyses were carried out using the SPSS version 25 for Mac (IBM SPSS Statistics, Armonk, NY, USA). Values were deemed significantly different when $P < 0.05$.

## RESULTS

The mean and SD in weight, length and diameter of the 10 logs was 5 ± 1.0 kg, 94 ± 0.9 cm and 9 ± 1.0 cm, respectively. The logs and fruiting bodies had a DW of 601 ± 41.4 and 109 ± 6.8 g/kg, respectively. There were on average 36 fruiting bodies per log.

The sawdust-to-shiitake TF was numerically lower ($P > 0.05$) than both the sapwood-to-shiitake TF and heartwood-to-shiitake TF. The mean concentration of $^{133}$Cs in the sapwood and heartwood was lower ($P = 0.043$ and $P = 0.037$, respectively) than in the bark (Table 1). Along the length of the logs, $^{133}$Cs concentrations in sawdust (Fig. 3A) and fruiting bodies (Fig. 3B) did not differ ($P > 0.05$) between cuts and discs, respectively. Stable cesium concentration in fruiting bodies collected from odd- (600 µg/kg DW) and even-numbered discs (581 µg/kg DW) did not differ ($P > 0.05$) (Fig. 3C) and the sawdust-to-shiitake TF from odd- (TF = 22.6) and even-numbered (TF = 23.1) cut positions did not differ ($P > 0.05$) (Fig. 4A). Stable cesium concentration in heartwood, sapwood, bark, sawdust and fruiting bodies were positively intercorrelated ($P < 0.05$; Table 2).

## DISCUSSION

Stem wood and bark have been reported to be chemically heterogeneous (Harju et al., 1996, 2002; Saarela et al., 2005), which would imply a rigorous sampling strategy is required when sampling logs for $^{133}$Cs. The current in-house methodology to ensure that wood and shiitake fruiting body samples for inorganic elemental analysis are representative involves cutting logs into nine 10 cm-discs, discarding the bark and mechanically breaking the wood into smaller pieces (i.e., crush and mill), and then compositing the wood from three discs to provide a total of three subsamples per log for $^{133}$Cs analysis (Fig. 1). Similarly, prior to cutting the log, all fruiting bodies are collected from each set of three discs, sliced and milled to provide three subsamples per log. Although this sampling method does provide a representative sample for analysis, it is very time-consuming and labor intensive to process samples from logs. In studies that require a larger number of logs to determine the log-to-shiitake TF of $^{133}$Cs for example, a more efficient methodology was required.

### Transfer factor of $^{133}$Cs between logs and shiitake

The average sawdust-to-shiitake TF of $^{133}$Cs was 23 (on a DW basis) and 4.2 (on a FW basis). To the best of our knowledge, this is the first report of the TF of $^{133}$Cs between logs and shiitake, although the TF of radiocesium was reported to be 0.43 between semi-dried logs (12% moisture content) and fresh fruiting bodies (Forestry Agency, 2012). One reason the TF of $^{133}$Cs is higher than the TF of radiocesium on a FW basis is because the log-to-shiitake TF of radiocesium was determined within 1 year of the nuclear accident when only the bark was contaminated (Mahara et al., 2014), whereas $^{133}$Cs was shown to be distributed through all parts of a log in this study (Table 1), and is likely to be more available for uptake. In addition, radiocesium released from the Fukushima Daiichi Nuclear Power Plant can be in the form of aerosol bound particles (Kaneyasu et al., 2012) and spherical Cs-bearing particles (Adachi et al., 2013) and thus their bioavailability to organisms may differ to $^{133}$Cs (Steinhauser, 2018).

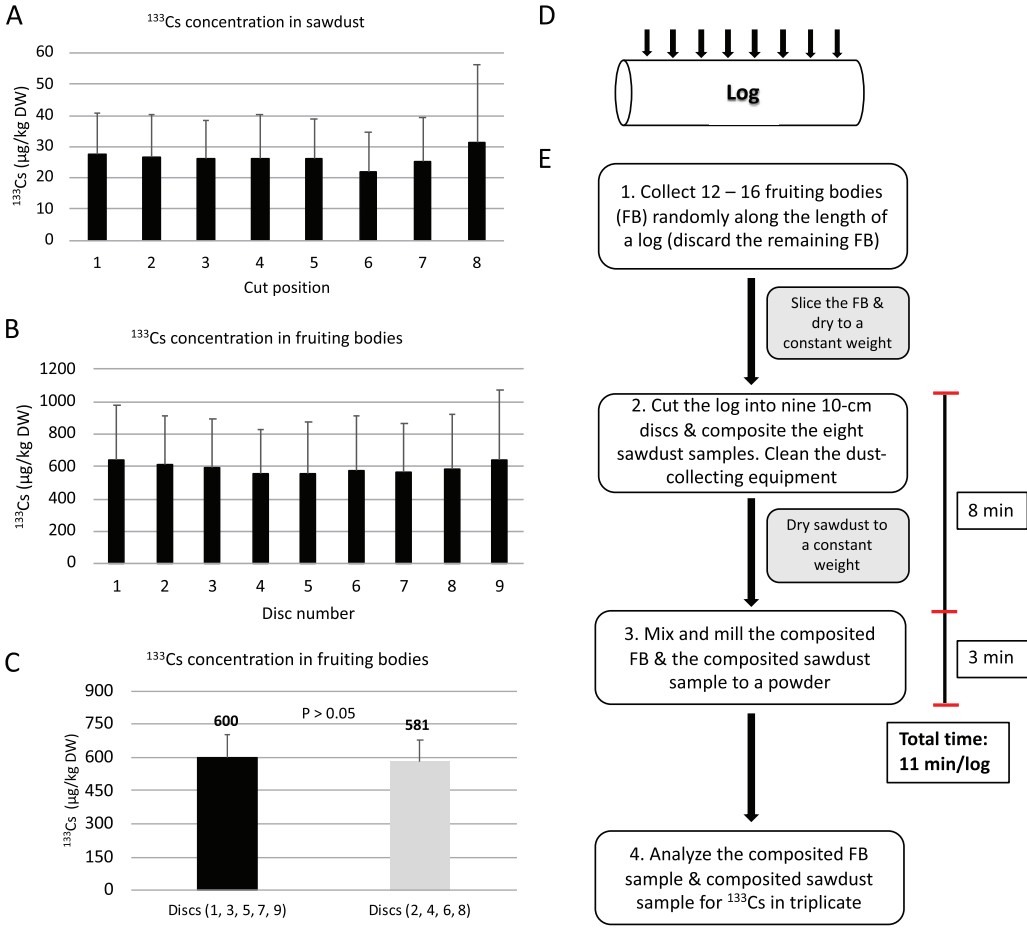

**Figure 3 Development of a whole-log sampling methodology (Method I).** (A) Stable cesium concentration (mean ± SD) in sawdust collected from eight positions along the length of the logs ($n = 10$ logs). The concentration in sawdust did not differ between cut positions (standard error of the mean (SEM), 1.4 µg/kg dry weight (DW); $P > 0.05$, one-way ANOVA). (B) Stable cesium concentration (mean ± SD) in fruiting bodies collected from nine discs along the length of the logs ($n = 10$ logs). The concentration in fruiting bodies did not differ between discs (SEM, 33.2 µg/kg DW; $P > 0.05$, one-way ANOVA). (C) Stable cesium concentration (mean ± SEM) in fruiting bodies collected from odd- and even-numbered discs ($n = 10$ logs); the data was analyzed with the paired sample $t$-test. (D) Locations of cut positions on logs proposed for Method I. (E) An overview of Method I including the approximate time required for one technician to collect a composited sawdust sample from one log (size = 20 × 90 cm); the total time stated excludes drying the sawdust and collecting and processing fruiting bodies.

In the current study, we found that the TF based on $^{133}$Cs concentration in heartwood, sapwood and sawdust was not significantly different. However, heartwood and sapwood had a lower concentration of $^{133}$Cs than bark, concurring with the findings of *Wang et al. (2018)* in 45–52-year-old oak trees. Even though the TF of $^{133}$Cs based on sawdust without bark was not directly compared to wood (i.e., sapwood and heartwood combined) in this study, the numerically lower TF value based on sawdust reflects the higher concentration of $^{133}$Cs in the bark. For shiitake cultivation, logs from trees aged *ca.* 15–25 years are used, and therefore it was important to confirm if $^{133}$Cs distribution between heartwood, sapwood and bark showed a similar trend as found in older oak trees.

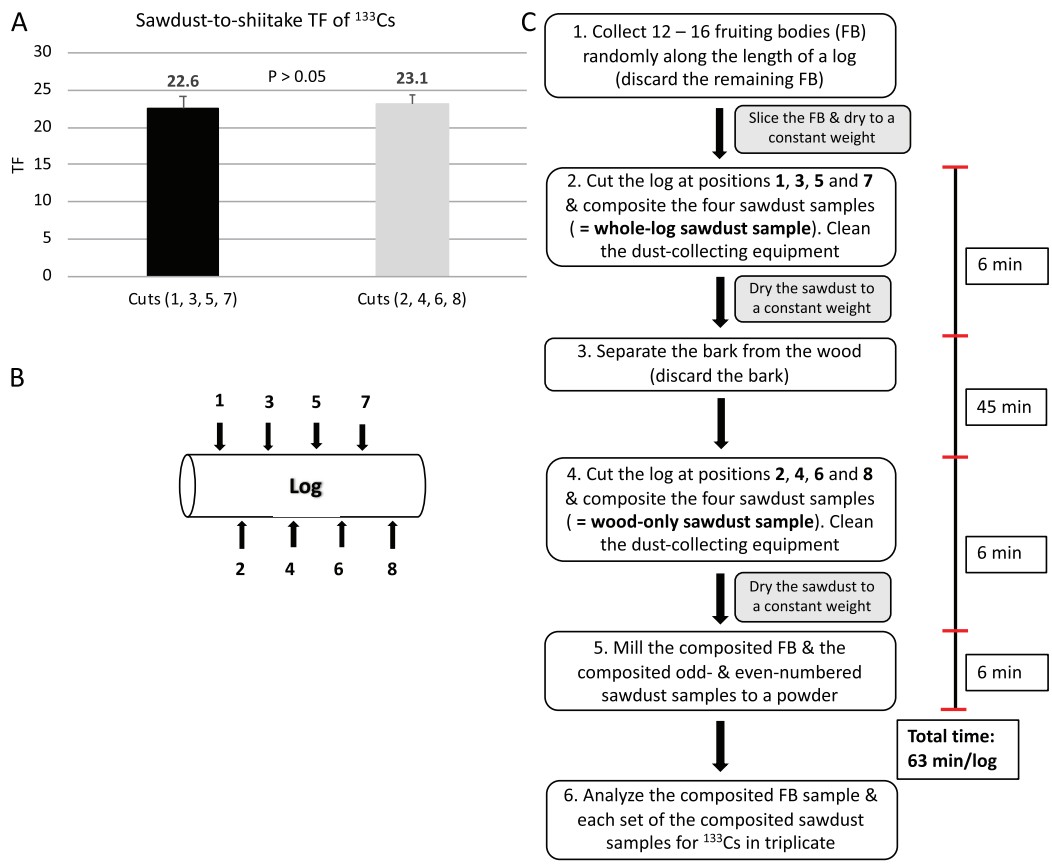

**Figure 4 Development of a combined whole-log and wood-only sampling methodology (Method II).**
(A) The transfer factor (TF) of $^{133}$Cs between the odd- and even-numbered whole-log sawdust samples. The TF was calculated based on Eq. (3) (see the section "Materials and Methods"). The data was analyzed with the paired sample $t$-test; error bars are standard error of the mean ($n = 10$ logs). (B) The location of cut positions on logs proposed for Method II. (C) An overview of Method II including the approximate time required for one technician to collect a composited whole-log and wood-only sawdust samples from one log (size = 20 × 90 cm); the total time stated excludes drying the sawdust samples and collecting and processing fruiting bodies.

**Table 2 Correlation of $^{133}$Cs concentration between oak log parts, sawdust and shiitake fruiting bodies ($n = 5$ logs).**

|  | Heartwood | Sapwood | Bark | Sawdust |
|---|---|---|---|---|
| Sapwood | 0.990** |  |  |  |
| Bark | 0.923* | 0.967** |  |  |
| Sawdust | 0.971** | 0.991** | 0.977** |  |
| Fruiting bodies | 0.952* | 0.978** | 0.987** | 0.971** |

Notes:
* $P < 0.05$.
** $P < 0.01$.

## Sampling of the whole log using sawdust

The concentration of $^{133}$Cs in sawdust and fruiting bodies along the length of the logs did not differ significantly and there were no unusually high concentrations in any one

sawdust sample or in fruiting bodies from any one disc in individual logs (Table 1). The natural variation in $^{133}$Cs concentration that was evident in sawdust (Fig. 3A) and fruiting bodies (Fig. 3B) can be overcome by collecting multiple samples per log during sampling. In the current study, the average concentration in eight sawdust samples was used to obtain an estimate of $^{133}$Cs concentration in each log. An alternative approach would be to collect a number of sawdust samples per log at pre-determined locations (eight was found to be satisfactory and convenient in the current study) and then mix these samples to produce one representative sample for the whole log. Comparing $^{133}$Cs concentration in fruiting bodies collected from odd- and even-numbered discs, we found no significant difference between these two sets of discs (Fig. 3C). With approximately four fruiting bodies on each 10-cm disc in the current study, our data suggest that collecting 16 fruiting bodies from the four even-numbered discs would be sufficiently representative of the shiitake crop growing on a log. However, if fruiting bodies were collected randomly along the length of a log to ensure at least one fruiting body per disk, 12 fruiting bodies would likely suffice. The methodology described above will be hereafter known as the whole-log sampling method (Method I) and has been illustrated in more detail in Figs. 3D–3E.

## Factors to consider when sampling logs and areas where further research is required

### Source of $^{133}$Cs in shiitake fruiting bodies

Based on what we can infer from the literature, bark ought to be included as part of the growth substrate of shiitake when determining the TF of $^{133}$Cs. Although *Tokimoto (2005)* reported that shiitake obtains its nutrients from the wood component of logs, *Matsumoto & Tokimoto (1992)* showed that a fruiting body may also acquire some of its nutrients from the inner bark. For example, the amount of some elements (e.g., Mg in the inner bark, Na, Fe and Cu in the inner bark and sapwood) increased beneath a fruiting body during the early stages of its development, and N, P and K decreased with the maturation of the fruiting body (*Matsumoto & Tokimoto, 1992*). There is also a notable decline in yield with each successive crop of mushrooms produced on logs (*Bratkovich, 1991*) which is likely due to one or more essential nutrients becoming limited (*Tokimoto, 2005*) and therefore, shiitake may be sourcing specific nutrients from both the wood and the bark depending on their availability and accessibility during fruiting body development. *Vane, Drage & Snape (2006)* found that shiitake will degrade the bark of oak logs if allowed to grow for an extended period of time.

*Yoshida & Muramatsu (1997)* and *Rühm et al. (1999)* reported the importance of identifying the soil layer from which certain species of fungi predominantly take up radiocesium, as it could lead to either an overestimation or underestimation of TFs (*Rühm et al., 1999*). In the case of shiitake, research is needed to determine what proportion of $^{133}$Cs in fruiting bodies is sourced from the wood and bark of logs, and if this proportion changes during consecutive shiitake harvests. The availability of $^{133}$Cs for uptake by shiitake from wood and bark would also need to be determined, as previously demonstrated between soil layers and mushrooms (*Baeza, Guillén & Bernedo, 2005*).

### Thickness of oak bark

Bark comprises 10–20% of woody vascular plants (*Vane, Drage & Snape, 2006*) and this percentage range can affect the accuracy of the sawdust-to-log TF of $^{133}$Cs when comparing logs cut from different tree species or trees of different age. In the current study, bark represented 19% of the total log DW but contained 38% of the total $^{133}$Cs on a DW basis. This discrepancy was due to a 2.6-fold higher $^{133}$Cs concentration in bark compared to wood. For researchers contemplating carrying out field trials relating to $^{133}$Cs or radiocesium, they should consider using logs with approximately the same proportion of bark (on a DW basis) across different treatments.

### Requirements of different professionals

When using $^{133}$Cs as a proxy element to understand the long-term behavior of radiocesium, plant and mushroom physiologist may prefer to use a TF of $^{133}$Cs based on wood only. Whereas in Japan, the government requires researchers in the food industry to report the TF of $^{133}$Cs based on the whole log. These preferences are likely based on either tradition or belief of where shiitake sources its $^{133}$Cs rather than from empirical evidence.

The three factors discussed above highlight the need for a more comprehensive and flexible sampling methodology. Below, we describe a second sampling methodology (Method II) to help satisfy the goals of different studies and professionals.

## Simultaneous sampling of the whole log and wood only using sawdust

Using data from the present study, we simulated sampling 10 logs using sawdust to determine the sawdust-to-shiitake TF between even- and odd-numbered cuts of the whole log (i.e., bark and wood) and found no significant difference (Fig. 4A). Although we did not repeat a similar analysis for wood-only sawdust samples, we believe that the significantly positive intercorrelation of $^{133}$Cs concentration between heartwood, sapwood and bark (Table 2) would indicate that the sawdust-to-shiitake TF between even- and odd-numbered cuts of wood-only samples would also not differ. Using this information, we showed that four sawdust samples per log can be used to represent either a whole-log or a wood-only sample. Method II involves collecting four sawdust samples per log at pre-determined locations (i.e., 1, 3, 5 and 7) and mixing these samples to produce one representative sample of the whole log. The bark is then removed from the discs and another four samples are collected at pre-determined locations (i.e., 2, 4, 6 and 8); these four samples are mixed to produce one representative sample of wood only (Figs. 4B and 4C). As per method I, 12–16 fruiting bodies should be collected randomly along the length of each log and composited. In Table 3, we provide some of the likely advantages and disadvantage of Methods I and II, as well as their potential uses in the field.

## Sampling logs efficiently

The current study did not compare the time and labor efficiencies obtained from collecting sawdust samples (Methods I and II) over wood samples (i.e., in-house sampling method) because it was obvious from the outset that major savings in both parameters would be found. However, from our experience sampling logs, we estimate that approximately

**Table 3 New sampling methodologies to determine the transfer factor (TF) of $^{133}$Cs between logs and shiitake.**

| Sampling methodology | Advantages[1] | Disadvantages[1] | Potential use[2] |
|---|---|---|---|
| Method I— whole log | • Lower sampling costs—no need to remove the bark before cutting or to break and crush the wood prior to milling to a powder<br>• Suitable for large sample sizes<br>• Sampling a larger number of logs ensures the sample mean TF ($\bar{x}$) will be closer to the population mean TF ($\mu$) (cf central limit theorem) | • On an individual log basis, the TF may be slightly underestimated because of a higher $^{133}$Cs concentration in bark[3] | • Surveys to determine the TF throughout a region/country<br>• Field trials |
| Method II— whole log & wood only | • Comprehensive—likely to satisfy the requirements of different studies and professionals who need to know the TF based on either the whole-log, wood-only samples or both materials<br>• From a food safety perspective, it would be important to know the upper limit of radiocesium transfer to shiitake based on wood-only samples[3] | • Higher sampling costs associated with removing the bark from each log<br>• Higher analytical costs associated with analyzing triplicate samples for both the whole log and wood only<br>• Less suitable for large sample sizes because of higher sampling and analytical costs | • Studies that require the TF to be reported based on both the whole-log and wood-only samples |

**Notes:**
[1] Between methods I and II and compared to the in-house sampling method using wood (Fig. 1).
[2] The number of logs that should be sampled will depend on the kind of information required by the investigation.
[3] The proportion of $^{133}$Cs in shiitake fruiting bodies originating from bark and wood is not known, nor its availability for uptake by shiitake from each part.

146 min is needed by one technician to sample a single 20 cm-diameter shiitake-inoculated log (i.e., the maximum size of log used to cultivate shiitake) using the in-house sampling method (Fig. 1). In contrast, it would take one technician *ca*. 11 and *ca*. 63 min to sample a log of similar size using Method I (Figs. 3D and 3E) and Method II (Figs. 4B and 4C), respectively. Collecting samples using sawdust eliminates the need to remove the bark (i.e., Method I) or to break and crush the wood into smaller pieces (i.e., Methods I and II). Method II is more time consuming than Method I because of the necessity to remove the bark before obtaining a wood-only sawdust sample. Methods I and II have an additional advantage over the in-house sampling method in that not all fruiting bodies on a log need to be collected and processed.

## CONCLUSIONS

Collecting multiple sawdust and fruiting body samples per log is a robust and efficient method to provide a representative sample to determine the TF of $^{133}$Cs from logs to shiitake. The use of sawdust will greatly reduce the time for sample collection and preparation, and this will facilitate sampling a larger number of logs in Eastern Japan to predict the future TF of radiocesium from contaminated logs to shiitake. To further refine the two proposed sampling methodologies discussed, it would be important to determine the proportion of $^{133}$Cs in shiitake fruiting bodies originating from bark and wood and its availability in each part.

## ACKNOWLEDGEMENTS

We would like to thank M. Takemura and M. Yoshikawa for their technical assistance in the laboratory, and to R. Sugita for help ordering equipment and supplies.

### Funding

This work was supported by the Japanese Society for the Promotion of Science (grant no. 18K05739). The funders had no role in study design, data collection and analysis, decision to publish, or preparation of the manuscript.

### Grant Disclosures

The following grant information was disclosed by the authors:
Japanese Society for the Promotion of Science: 18K05739.

### Competing Interests

The authors declare that they have no competing interests.

### Author Contributions

- Martin O'Brien conceived and designed the experiments, performed the experiments, analyzed the data, contributed reagents/materials/analysis tools, prepared figures and/or tables, authored or reviewed drafts of the paper, approved the final draft.
- Masakazu Hiraide conceived and designed the experiments, performed the experiments, contributed reagents/materials/analysis tools, authored or reviewed drafts of the paper, approved the final draft.
- Yoshimi Ohmae performed the experiments, authored or reviewed drafts of the paper, approved the final draft.
- Naoto Nihei conceived and designed the experiments, authored or reviewed drafts of the paper, approved the final draft.
- Satoru Miura conceived and designed the experiments, authored or reviewed drafts of the paper, approved the final draft.
- Keitaro Tanoi conceived and designed the experiments, contributed reagents/materials/analysis tools, authored or reviewed drafts of the paper, approved the final draft.

### Field Study Permissions

The following information was supplied relating to field study approvals (i.e., approving body and any reference numbers):

We obtained permission from Mr Atsuhiko Iizumi, the owner of Nakanokinokoen mushroom farm, to collect samples on his property.

### Data Availability

All raw data are available in the Supplemental File.

### Supplemental Information

Supplemental information for this article can be found online at http://dx.doi.org/10.7717/peerj.7825#supplemental-information.

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
