# Peer review of "Efficient sampling of shiitake-inoculated oak logs to determine the log-to-mushroom transfer factor of stable cesium"

_PeerJ, doi:10.7717/peerj.7825_

## Round 0.1 · original submission · Major Revisions

Please respond to the reviewer comments and revise your manuscript as appropriate. One of the reviewers has also included an annotated manuscript (attached).

Reviewer 1 ·

Basic reporting

The paper is written in clear standard English. I urge the authors to switch the notation of nuclides such as Cs-133 (the old type-writer notation) to 133Cs (superscripted 133). This refers to the text and the figures and tables! Also, it is not good style to start a sentence with a chemical symbol, such as in L 42.

Experimental design

The research question is well defined and original. The authors used apt methods to answer the research question. Results may be repeated based on their description.

L 160 ff. Could the authors comment on the QC procedures they applied during analysis, please?

Validity of the findings

Data are presented and discussed in a convincing fashion. For further comments, see below:

Additional comments

L 59. What is the regulatory limit for dried mushrooms?
L103. The latin name should appear upon first appearance in the intriduction
L 105. What is the rationale/justification for the sawdust samples? Are they to represent the integral Cs content across the cross section?
L 129. Is it known how far/how deep the mycelium grows into the log. DOes the 10 cm disk actually represent the mushroom's source of nutrients? I feel there should be some knowledge on this crucial question available in literature.

Speaking of literature:
Reading the paper I had the feeling that the article was not yet sufficiently embedded in the state of knowledge. May I suggest expanding aspects of the literature:

Mechanisms about uptake and applications: DOI.org/10.1007/s11356-017-9826-3
Ectomycorrhizal vs saprophytic mushrooms https://doi.org/10.1007/s11356-016-7027-0 and https://doi.org/10.1007/s11356-017-0538-5
Dose aspects: https://doi.org/10.1007/s11356-019-05376-8 and https://doi.org/10.1016/j.foodchem.2013.12.083
As well as current literature on the monitoring of mushrooms in Japan.

L163 like -> likely

L166ff. This sentence does not make any sense. Concentrations were low? I presume the deviation between the analytical results (=concentrations) was low?

L172. Purity of the HNO3, please

L173 w -> W

L174 µl should be µm?

L 177. ppb should no longer be used (µg/kg)

L 219/220: Dry weight has a unit of g/kg?! This sentence is confusing. What was measured? Density?

·

Basic reporting

Clearly written with very good English

Experimental design

Concerns over the sawdust method in the ability to separate material from heartwood, sapwood and bark. This is an essential component of the manuscript and has to be addressed (see comments below and on the manuscript).

Validity of the findings

Given the above criticism the data have been appropriately analyzed and are presented clearly

Additional comments

This is a relevant manuscript in relation to problems of radioactive fallout and particularly concentrates on transfer factors into food items. The manuscript I well written and the English is good.
I have made a few comments on the manuscript itself with the main criticism being lack of detail in the methodology. Unless I have missed a major comment, I cannot see how the collection of sawdust from sawing intact logs can allow the separation of heartwood, sapwood and bark for analysis of Cs content. The discussion of the role of each log component to the accumulation of Cs in the mushrooms is a major part of this manuscript and the methods need to be clearly stated.
Initially in the manuscript, bark was dismissed as a source of Cs for mushrooms. However data suggested that bark may play a more important role as a Cs source. This seems to me to call for a set of experiments where mushrooms are experimentally grown on each log component and Cs transfer measured.
Based on these comments I would suggest the manuscript need some major revision before it could be accepted for publication.

Reviewer 3 ·

Basic reporting

This article, describing a more time efficient sampling of shiitake logs via sawdust sampling, is generally well-written and holds interesting aspects on the time efficient sampling of logs. I am sure, that the shiitake growers will profit of an faster sampling method. The presentation of the Cs-133 concentration in log10 is rather uncommon, especially in stastical evaluation of the results, even though the concentrations are accessible in the Supplementary information. I would strongly recommend using the concentrations without log10 and with that a new statistical evaluation on these values. The biological variances for Cs-133 are clearer seen and the statistics do not seem smoothed.

Experimental design

The experimental design is well thought-out. In the section Cs-133 analyses, I am missing the used standards for Cs analyses and the purity used for the chemicals. I generally think that the use of log10 concentrations even when stated in Data analysis are not practical as biological variances cannot be seen, and the concentrations do not vary so strongly, the data analyses benefits from its use. Furthermore, it hinders the easy comparablibilty with other studies.

Validity of the findings

A faster approach for Cs-133 determination in sawdust from logs is described with 2 new and faster methods and compared with the commonly whole log used method. As stated before, the use of log10 concentration (Table 1) is uncommon, especially the calculation of mean values and standard deviation from log10 values. Preferably use the concentration values. Especially as the transfer factors are calculated with the non-log values. In regards to the log10 values figure 2 does not show any significant differences neither in values nor errors, a presentation of the non-log values would be better, as the concentration do not vary over that many magnitudes.
Please consider revising the statistics for non-log values.
The concentration values in the Supplementary Information further need measurement uncertainties. Furthermore, please only give significant decimal places (Table 1).
In several lines, you underlined, that the new sawdust method is faster than the whole log sampling, please give a value and comparison for the time improvement of the techniques.
Line 247-254 you found significant higher transfer factors for non radioactive cesium than for radiocesium, a possible effect could be the bioavailibility of cesium, whereas stable cesium is found in nature in mostly ionic form the uptake for organisms as potassium analogon is easier. Radiocesium is often also released aerosol bound (steinhauser, JRNC, 2018) or even in glass particles in Fukushima NPP accident (Adachi et. al., Scientific Reports, 2013). Please consider adding to the discussion.

Additional comments

The article shows improvement in sampling and measurement of Cs-133 in shiitake logs with sawdust sampling. The article is well written and the experiment well thought. I would recommend using non-log values for Cs-133 concentrations for comparability and statistics.

---

## Round 0.2 · accepted · Accept

Thank you for your efforts to respond to reviewer comments and revise your manuscript.

·

Basic reporting

no comment

Experimental design

Authors have addressed my concerns

Validity of the findings

Modifications have clarified interpretation of the results

Additional comments

Authors have satisfactorily addressed the concerns I had with the initial manuscript and their modifications have significantly improved the clarity of the manuscript.

Reviewer 3 ·

Basic reporting

The revised manuscript shows a clear and consistent experimental design and results.

Experimental design

The additional discussion on the time efficiency adds relevant information to the aim and findings of the study.

Validity of the findings

The non log concentration were changed accordingly and gives a better insight on the data set.

Additional comments

I found only some small mistakes in:
L. 168 Cs-133 analyses it should be analysis
L. 178 the unit is still the old one with log10, please change to only µg/kg